# The Multifaceted Roles of Mast Cells in Immune Homeostasis, Infections and Cancers

**DOI:** 10.3390/ijms23042249

**Published:** 2022-02-17

**Authors:** Anna Sobiepanek, Łukasz Kuryk, Mariangela Garofalo, Sandeep Kumar, Joanna Baran, Paulina Musolf, Frank Siebenhaar, Joachim Wilhelm Fluhr, Tomasz Kobiela, Roberto Plasenzotti, Karl Kuchler, Monika Staniszewska

**Affiliations:** 1Faculty of Chemistry, Warsaw University of Technology, Noakowskiego 3, 00-664 Warsaw, Poland; asobiepanek@ch.pw.edu.pl (A.S.); joanna.baran.dok@pw.edu.pl (J.B.); pimusolf@gmail.com (P.M.); tomasz.kobiela@pw.edu.pl (T.K.); 2National Institute of Public Health NIH—National Institute of Research, 00-791 Warsaw, Poland; lkuryk@pzh.gov.pl; 3Clinical Science, Targovax Oy, Lars Sonckin kaari 14, 02600 Espoo, Finland; sandeep.kumar@targovax.com; 4Department of Pharmaceutical and Pharmacological Sciences, University of Padova, Via F. Marzolo 5, 35131 Padova, Italy; mariangela.garofalo@unipd.it; 5Institute of Allergology, Charité - Universitätsmedizin Berlin, Corporate Member of Freie Universität Berlin, Humboldt-Universität zu Berlin, and Berlin Institute of Health, 10117 Berlin, Germany; frank.siebenhaar@charite.de (F.S.); joachim.fluhr@charite.de (J.W.F.); 6Fraunhofer Institute for Translational Medicine and Pharmacology ITMP, Allergology and Immunology, 12203 Berlin, Germany; 7Department of Biomedical Research, Medical University of Vienna, Währingergürtel 18-20, 1090 Vienna, Austria; roberto.plasenzotti@meduniwien.ac.at; 8Max Perutz Labs Vienna, Center for Medical Biochemistry, Medical University of Vienna, Campus Vienna Biocenter, Dr. Bohr-Gasse 9/2, 1030 Vienna, Austria; karl.kuchler@meduniwien.ac.at; 9Centre for Advanced Materials and Technologies, Warsaw University of Technology, Poleczki 19, 02-822 Warsaw, Poland

**Keywords:** mast cells, receptors, degranulation, inflammation, allergic response, anti-pathogen defense, tumorigenesis, immunotherapy

## Abstract

Mast cells (MCs) play important roles in normal immune responses and pathological states. The location of MCs on the boundaries between tissues and the external environment, including gut mucosal surfaces, lungs, skin, and around blood vessels, suggests a multitude of immunological functions. Thus, MCs are pivotal for host defense against different antigens, including allergens and microbial pathogens. MCs can produce and respond to physiological mediators and chemokines to modulate inflammation. As long-lived, tissue-resident cells, MCs indeed mediate acute inflammatory responses such as those evident in allergic reactions. Furthermore, MCs participate in innate and adaptive immune responses to bacteria, viruses, fungi, and parasites. The control of MC activation or stabilization is a powerful tool in regulating tissue homeostasis and pathogen clearance. Moreover, MCs contribute to maintaining the homeostatic equilibrium between host and resident microbiota, and they engage in crosstalk between the resident and recruited hematopoietic cells. In this review, we provide a comprehensive overview of the functions of MCs in health and disease. Further, we discuss how mouse models of MC deficiency have become useful tools for establishing MCs as a potential cellular target for treating inflammatory disorders.

## 1. Introduction

Mast cells (MCs) constitute a major component of innate immunity due to their distribution in the superficial dermis proximal to blood and lymphatic vessels, nerve fibers, and in muscle tissues [1]. MCs emerge from the multipotential hematopoietic stem cells (MHSC) as progenitors (MCp) during hematopoiesis regulated by the transcriptional factors (GATA-1, GATA-2, and PU.1) and the microphthalmia transcription factor (MITF). After leaving hematopoietic niches, MCs migrate via peripheral blood and invade connective or mucosal tissues, where they complete differentiation [2].

MCs are important for the recognition and clearance of microbial pathogens and the initiation of adaptive immune responses during microbial infection, as well as for the activation of the cell–cell crosstalk [3,4]. Moreover, MCs play immunopathological roles in a number of inflammatory disorders since they release chemokines that recruit leukocyte subtypes harmful for host tissues. In autoimmune diseases (rheumatoid arthritis), chemokines (CXCL12, CCL2, CCL3, CCL4, and CCL5) are implicated in tissue destruction [5,6,7].

In this review, we comprehensively discuss the research into roles of MCs in immune homeostasis as well as their multiple roles in the host defense against pathogens, whereby MCs act as sensors and effectors of responses, as controllers of infection and modulators of local inflammation. We discuss the significance of MCs in complex interactions with other components of the mucus layers, tissue-resident microbiota, and epithelial cells, and their role in the recruitment of other innate and adaptive immune cells. Furthermore, we discuss the functions of MCs in maintaining the equilibria and homeostasis between host and microbiota, including the crosstalk between resident hematopoietic cells. Finally, we focus on the induction of MC degranulation, which is often associated with microbial invasion and IgE-receptor aggregation, and we review their role during allergic and autoimmune inflammation.

## 2. Mast Cell Pattern Recognition Receptors

Mast cells are usually activated via the surface-type receptors, including Toll-like receptors (TLRs), major histocompatibility complexes MHCs (class I and II), complement receptors (CR1-5, C3aR, and C5aR), cytokine receptors (IL-1R, IL-3R, IL-10R IL-12R, INFγR, TGFβR, and MRGPRX2), and chemokine receptors (CCR1, CCR3, CCR4, CCR5, CCR7, CXCR1, CXCR2, CXCR3, CXCR4, CXCR6, and CXCR1) [8,9]. MCs also express receptors for histamine (H1/H2/H3), adhesion (ICAM-1, VCAM, VLA4, and CD226), and co-stimulatory molecules (CD40L, OX40L, CD86, and CD80) that fine-tune T and B cells by either enhancing or attenuating their responses [10,11]. 

The non-IgE-dependent activation of MCs involves, as follows: TLRs, the NOD-like receptor (NLR), the cluster of differentiation 48 (CD-48), and leucine-rich repeat proteins (NRRP) present on MCs [12]. TLR2 and Dectin-1 (CLR family) recognize various pathogens (Figure 1) by pathogen-associated molecular patterns (PAMPs), such as: β-1,3-glucan, phospholipomannan, lipopolysaccharide (LPS), and double-stranded RNA (dsRNA). Dectin-1 and TLRs trigger the family of low-molecular-weight GDP/GTP-binding guanine triphosphate (Ras) and GTPase/serine/threonine-specific kinase isoform (Raf) [13,14]. The complement receptor CD-48 is directly activated by the FimH adhesion protein expressed by many bacteria [15]. Moreover, TLR4 and TLR6 are involved in the recognition of mannoproteins, and they cooperate with Dectin-1 to boost cytokine release in response to β-glucans [16]. These mechanisms also activate signaling through the spleen tyrosine kinase (Syk) pathway, leading to the activation of the nuclear factor kappa-light-chain-enhancer of activated B cells (NF-kB) [7,17]. Moreover, mROS (mitochondrial reactive oxygen species) production by activation of the redox-sensing protein kinase C (PKC) occurs downstream of TLR signaling after pathogen infections. The activation of Bruton’s tyrosine kinase (BTK) is involved in several signaling pathways, including TLR, Fc receptor, and FcεRI [18,19,20]. The activated BTK interacts with numerous proteins, including C-terminal Toll/interleukin 1 (TIR) homology domain, the adaptor protein (TIRAP), TIR domain including IFN-β (TRIF) and TRIF-related adaptor molecules (TRAM), as well as myeloid differentiation factor 88 (MyD88) [21]. The receptor-mediated activation of phosphoinositide-specific phospholipase C (PLC) triggers hydrolysis of phosphatidylinositol 4,5-bisphosphate (PIP2) to yield the secondary messenger inositol 1,4,5-triphosphate (IP3) and diacylglycerol (DAG). DAG activates and recruits protein kinase (PKC) to the membrane, along with activating the transcription and pro-inflammatory cytokine production in myeloid cells [22,23]. NLR inflammasomes, which interact with cytoplasmic microbial products, trigger maturation and secretion of pro-inflammatory IL-1β as well as IL-18 and IL-33, effectively guarding immune responses [24,25,26]. Dectin-1 is a crucial CLR receptor mediating recognition and phagocytosis as well as the killing of fungal pathogens [16,27]. Of note, as another CLR in MCs (Mincle), it contributes to immunity through Syk activation, which is triggered by association with FcεRI [17,28]. A synergistic activation between TLR2 and FcεRI (which captures monomeric IgE) occurs in MCs, resulting in the increased release of inflammatory cytokines [13,19,21]. Histamine and adenosine bind to the G-protein-coupled H1-H4 and A1 receptors, respectively, to trigger β-endorphin secretion after mechanical stimulation of MCs [9,22,29]. Bacterial infections induce the expression of additional monoamine neurotransmitters, including endothelin-1 and neurotensin-1, which signal through G-protein-coupled receptors (GPCR), such as 5-HT [9,27,30]. Finally, C5a controls histamine release by MCs via the complement C5a receptor [31].

## 3. Mast Cell Secretory Granules

MC secretory granules emerge from small pro-granules, followed by clathrin-coated vesicles budding off from the trans-Golgi network. Next, the fusion of pro-granules occurs, and finally, immature granules undergo maturation. This process is regulated by secretogranin III, GTPase Rab5, and ATPase (V-ATPase) pumping cytosolic protons into the granules to acidify the granule lumen [23]. The electron-dense granules contain numerous bioactive substances, including proteases categorized into three classes, namely two types of serine proteases known as tryptases (mMCP6, -7, -11, and mTMT—transmembrane tryptase) and chymases (mMCP-1,-2,-4,-5, and -9), as well as the metalloprotease carboxypeptidase A (mMC-CPA3) [32,33]. The proteases are sorted to the pro-granules from the trans-Golgi network [34]. Human MCs are categorized as tryptase-containing MCs (MCTs) or as tryptase- and chymase-containing MCTCs [35]. The murine chymase locus on chromosome 14q11.2 encodes six proteases, including mMCP1, mMCP2, mMCP4, mMCP5, mMCP9, and mMCP10. Moreover, the granzyme-related gene Mcpt8 encodes mMCP8 and six functional granzyme genes [36]. Human chymase locus 14C1/2 encodes four genes, such as chymase CMA1, cathepsin G, granzyme B, and granzyme H [37]. The tryptase locus encodes 13 trypsin-like serine proteases and 9 proteases on mouse chromosome 17A3.3 and on human chromosome 16p13.3, respectively [38]. Tryptases and chymases are bound to heparin in the granules of CTMC (connective tissue mast cells). Chymases are usually bound to chondroitin sulphate in MMC (mucosal mast cells). Generally, proteases induce inflammatory cell mobilization to an infection site [39,40]. The tryptase-β2 MCP6 appears to be involved in the host response to peritonitis by recruiting neutrophils into the peritoneal cavity [41]. Of note, evidence indicates that proteases can directly kill bacteria [27]. Moreover, lack of MCP6 hampers the control of chronic parasitic infection due to impaired recruitment eosinophils. Chymases act as chemoattractants that can regulate the neutrophil function in-vitro and in-vivo [42]. They are responsible for limiting the harmful effects of endogenous mediators, such as neurotensin and endothelin-1, which cause hypotension and vasoconstriction in septic responses. These peptides can be downregulated by CPA (carboxypeptidase) and cleavaged by neurolysin [43]. Patients with MC activation symptoms show elevated tryptase levels caused by hereditary autosomal dominant α-tryptasemia (HαT), which is characterized by the inheritance of multiple copies of the TPSAB1 gene encoding α-tryptase [44]. Furthermore, MC granules contain heparin sulfate, serotonin, ATP, and lysosomal enzymes such as β-hexosaminidase, the latter cleaving the terminal linked-N-acetylhexosamine residues in N-acetyl-β-hexosaminides in granules rather than in MC lysosomes. Moreover, the enzyme is crucial for bacterial and fungal defense, though without driving allergic responses [1,45].

The major component of MC granules, histamine is released to the extracellular space by degranulation, which is subject to regulation by a physical/mechanical stimulation of MCs. The biogenic amine histamine enters granules through the vesicular monoamine transporter VMAT2. Of note, the MC circadian gene clock controls cytosolic histamine levels synthesized by histidine decarboxylase before VMAT2-mediated sequestration into granules, and the release to the extracellular environment upon stimuli [46,47,48]. The elevation of the cytosolic Ca^2+^ level is the key signal for MC degranulation and cytokine production. The OCT3 target gene clock regulates histamine release from MCs, thus determining plasma histamine levels, critical for the amplitude of allergic reactions. Another degranulation regulator, the vesicular aminergic-associated transporter (SLC10A4), affects intra/extracellular ATP levels by directly or indirectly facilitating the transport of ATP into the MC granules. Hence, these control elements impact granule fusion with the plasma membrane as well as the release of granule-associated mediators that lead to powerful inflammatory reactions [46,48,49]. 

An invasion by microbes or exposure to bacterial and fungal pathogen-associated patterns (PAMPs) or toxins generated by infecting microbes trigger MC degranulation [50]. Furthermore, mechanical or heat stimulation of the transient receptor of potential cation channel subfamily V, -member 2 (TRPV2) in the plasma membrane elicits MC degranulation to trigger acupoint signals in analgesia. There is a relationship between MC activation and local adenosine signaling triggered after the deformation of the collagen fiber in the acupoint [29]. Under stress conditions, MCs release ATP-inducing ectonucleotidases present on the surface of MCs to hydrolyze ATP in order to liberate adenosine diphosphate, monophosphate, and adenosine [51]. Furthermore, a positive feedback loop in activated MCs releases adenosine. Extracellular adenosine binds to A3 receptors in MCs to induce their degranulation via chemical activation, thereby initiating an additional release of adenosine and histamine [52]. 

## 4. Mast Cell Interaction with Other Mammalian Cells

MCs participate in a broad spectrum of physiological processes implicated in normal tissue homeostasis (e.g., wound healing, tissue repair, and skin barrier homeostasis), as well as in the immune surveillance of pathological homeostasis or infection conditions, including allergic diseases, angiogenesis, and tumor initiation and progression. Hence, MCs must communicate within tissues with many other cell types via direct cell–cell interactions engaging cell–extracellular matrix (ECM) interactions to orchestrate balanced responses in concert with both immune and non-immune cells. The indirect interaction of MCs with other cells involves the release of specific mediators [9,53], but direct interactions are possible through a ligand–receptor binding (Table 1; Figure 2). Especially at inflamed sites, MCs control the local inflammatory responses, whereby the release of histamine, tryptase, and leukotrienes recruit effector cells to inflamed areas. Interactions with specific receptors may help to achieve homeostasis of the immune system [54].

A tight interaction occurs between MCs and lymphocytes. This communication provides a co-stimulation engaging bidirectional and reciprocal signals in regulatory or modulatory roles. This occurs because, under certain circumstances, MCs can serve as antigen-presenting cells (APCs) for instructing T cells [89]. Examples of MC-T cell’s interactions are listed in Table 2. The direct recognition of surface receptors on the T cells with antigens presented by MCs stimulates their degranulation. The release of several mediators induces lymphocyte activation, proliferation, and survival [90]. Furthermore, the MC-released mediators, such as TNF-α and IL-4, are capable of inducing the T cell adhesion to human endothelial cells or fibroblasts by increasing the expression of cell-surface adhesion molecules, including the intercellular adhesion molecule (ICAM)-1, endothelial-leukocyte adhesion molecule (ELAM)-1, and the vascular cell adhesive molecule (VCAM)-1) [91]. Of note, such co-stimulation pathways may offer therapeutic targets for treating certain autoimmune diseases, such as type 1 diabetes mellitus in infancy, hypothyroidism, and Addison’s disease [92].

MC progenitors reach the endothelium and migrate via vessels to travel to distant tissue sites (including inflamed areas). Thus, MCs express several families of surface adhesion molecules, such as the integrins very late antigen (VLA)-4 and VLA-5 for binding to extracellular components, including fibronectin (FN), laminin (LN), vitronectin (VN), and collagen (COL) types present in connective tissues. MCs determine their presence in specific tissues by modifying the expression of these receptors [64]. Additionally, the MC adhesion to endothelial cells and ECM ligands is upregulated by interaction with the T cells, which is accompanied by the mitogen-activated protein (MAP) kinase activation, and further enhanced by the FcεRI-mediated degranulation of MCs. In turn, the release of soluble proteases controls ECM degradation during inflammation and tissue remodeling [74,117]. On the other hand, some MC mediators may also stimulate fibroblasts to induce the synthesis of ECM proteins during tissue remodeling. For example, not only does the tryptase-induced activation of the PAR2 receptor (protease activated receptor 2), also known as coagulation factor II (thrombin) receptor-like 1 (F2RL1) or GPCR11 present on fibroblasts, increase migration and proliferation of fibroblasts, but it also enhances the production of type I pro-collagen [39]. However, tryptases induce angiogenesis by releasing ECM-bound angiogenic factors, including cytokines and metalloproteinases (MMPs). At the same time, activation of several adenosine receptors expressed on MCs may stimulate the synthesis of the classical proangiogenic factors (PFs). For instance, activation of the A2A receptor stimulates IL-8 and vascular endothelial growth factor (VEGF), along with the activation of A3 receptor-induced expression of angiopoietin-2 (Ang-2) [69,74,83].

The location of MCs in the perivascular space, and in particular in close proximity to neurons, also suggests functional MC-neuronal interactions (Conti et al. 2020). This physical association is facilitated by N-cadherin, nectin-3, and cell adhesion molecule 1 (CADM1), also known as synaptic cellular adhesion molecule 1 (SynCAM 1); or nectin-like molecule 2 on MCs. CADM1 appears to provide a microenvironment enhancing MCs’ sensitivity to neuronal signals [53]. The generation of the protective innate immune responses in the tissue results from adhesion junctions formed between MCs and neurons, but also to other types of skin immune cells, such as dendritic cells (DCs). DCs are key players in the immune system at the intersection of innate and adaptive immunity, as DCs are professional APCs. DCs interact directly with IgE/antigen-activated MCs via the lymphocyte function-associated antigen 1 (LFA-1) and very late antigen 4 (VLA-4) integrins, leading to cytoplasmic reorientation, directed degranulation of MCs, and cytokine signaling towards DCs. In this contact-mediated signaling, the IgE-bound antigens are transferred from MCs to DCs. By contrast, in the contact-independent mechanisms, the MC-derived TNF matures DCs and downregulates the expression of β-chemokines, such as CCL2 and CCL3. The MC-derived histamine can block IL-12 or induce the IL-6 and INF-γ release from DCs. In particular, the MC-derived mediators TNF-α and IL-1β induce the migration of Langerhans cells, which happens upon TLR2 or TLR7 activation by cognate ligands [40,49,67]. Although the nature of MC-neutrophil interactions still remains to be clarified, an association between these cells has been noted in a series of gastric carcinomas. MC-derived TNF-α recruits neutrophils to infection sites or other pathological conditions, such as wound repair after tissue damage [5,7,81]. MC-derived tryptases and chymases are pro-apoptotic, and both potentiate the pro-apoptotic action of histamine. Indeed, degranulation of MC induces apoptosis in vascular smooth muscle cells, endothelial cells, and macrophages [56,118]. MCs are also key effectors in the pathogenesis of asthma via their association with the airway smooth muscle (ASM). MCs and eosinophils are relevant in allergic diseases, as well as in inflammatory and neoplastic disorders. In allergic disorders, MCs and eosinophils form the ‘allergic effector unit’; in tumors, these cells are found in close proximity with each other, suggesting that they modulate their functions in a bidirectional way. Eosinophils are activated via MC-released mediators, such as IL-5, stem cell factor, histamine, platelet-activating factor (PAF), prostaglandin (PGD2), and cysteinyl leukotrienes, whereas vascular endothelial growth factors (VEGFs) activate eosinophils. On the other hand, eosinophil cationic proteins, major basic protein (MBP), nerve growth factor, and VEGFs released by activated eosinophils modulate the MC function [56,67,87]. Moreover, the interaction between MC and fibroblasts is bidirectional in health and disease. They interact due to their anatomical proximity in connective tissue. MCs’ survival and differentiation are regulated by fibroblasts, whereas biochemical changes in fibroblasts leading to fibrosis are induced by MCs [119]. MCs are implicated in cutaneous repair processes by producing fibroblast growth factors (FGF), and MCs stimulate the production of FGF in dermal fibroblasts. It was shown that both dermal and leukemic MCs express FGF-2/7 and heparin-binding epidermal growth factor, whereas dermal MCs additionally express FGF-10. Of note, tryptase-dependent lung fibroblast proliferation is mediated via the PAR-2 receptor, while tryptase acts on dermal fibroblasts via the control of the FGF-2 synthesis, which, in turn, promotes fibroblast proliferation [120,121,122].

## 5. Mast Cells in Tumor Microenvironments—From Basics to Clinics

In the late 1870s, Paul Ehrlich described the existence of MCs in tumor microenvironments (TME). Since then, compelling evidence has confirmed the presence of MCs within cancer tissues, now referred to as tumor-associated MCs (TAMCs) [56,74]. Remarkably, TAMCs can adopt ‘Dr. Jekyll and Mr. Hyde’ phenotypes, as they can be both pro- and anti-tumorigenic, or just neutral bystanders [93,123] (Table 2).

MCs can promote tumor development by (i) disrupting stroma-epithelial communication, (ii) mediating tumor vascularization, and (iii) inducing the release of various growth factors such as stem cell factor (SCF) [124]. Elevated numbers of MCs in tumor mass correlate with a negative prognosis, increased metastasis, and poor survival in several tumor types, including melanoma [125,126], prostate cancer [106], pancreatic adenocarcinoma [70], squamous cell carcinoma [112] and Hodgkin lymphoma [127]. MC accumulation may occur due to the chemotactic activity elicited by RANTES (regulated upon activation, normally T-expressed) or by the chemokine monocyte chemoattractant protein 1 (MCP-1). In addition, histamine can also drive cancer proliferation through H1 receptors and simultaneously suppress the host immune defense through another H2 receptor [60,128]. Of note, both histamine receptor-binding sites are present on the surface of tumor cells. MC mediators may also promote brain metastases due to changes in the permeability of the blood–brain barrier (BBB). Furthermore, acute stress also enhances BBB permeability in a MC-dependent manner [82,129].

Cancer cells release inflammatory mediators as well as pro-angiogenic factors, including fibroblast growth factor 2 (FGF-2), SCF, and nerve growth factor (NGF), thus activating vascular epithermal growth factor (VEGF) and SCF/c-KIT signaling among others. These events result in the maturation, migration, and survival of MCs. The tissues surrounding tumors contribute to MC infiltration of the TME via SCF chemotaxis, pro-angiogenic mediators, proteases, and growth factors, all in all promoting tumor progression. Furthermore, MCs release FGF-2 and VEGF to induce angiogenesis [8,64,93]. Therefore, MCs are often in proximity to the CD31+ cells and blood vessels [130]. MCs also release tryptase, which contributes to extracellular matrix degradation and vascularization, thus mediating angiogenesis and tumor growth as well as metastasis [33,131]. MCs can also facilitate tumor progression by modulating TME and the development of anticancer drug resistance. MCs can also promote cancer development by secreting specific cytokines although without degranulation (differential release, intragranular activation). For example, release of inflammatory IL-6 can proceed independently of histamine [64,132,133].

In contrast, MCs can mediate anti-cancer responses. For instance, the MC accumulation in a breast tumor and the surrounding lymph nodes can mediate tumor regression or even rejection. Furthermore, MCs in adenocarcinomas can release various cytokines and proteolytic enzymes, inhibiting cancer cell growth, including the anticoagulant and angiogenic heparin. The MC tryptase can also promote protease-activated receptors, such as PAR-1 or PAR-2, stimulated by thrombin and trypsin [93]. Protamine, which neutralizes the anticoagulant properties of heparin, can trigger blood vessel thrombosis in the tumor mass. The heparin-related sulfated glycosaminoglycans (GAG) and heparan sulfate (HS) interact with modifiers on the surface of MCs in cancer cell type [134,135]. Although cancer cells can metastasize by binding to chondroitin sulfate (CS), the exogenous application of CS inhibits metastasis of ovarian carcinoma [136]. In addition, HS proteoglycans can negatively affect heparan binding to cells, thereby preventing new vesicles within a tumor [76]. MCs can also provide pro-inflammatory responses and contribute to the impairment of tumor development through innate and adaptive immune mechanisms [97,137]. Of note, MCs can mediate the infiltration of various immune cells into tumors, thus affecting the clinical outcome.

## 6. The Role of Mast Cells in Tumor Progression and Regression—Preclinical Studies

Animal models with specific mutations or deletions in the c-KIT gene yielded new knowledge of MCs’ biology and their enhancing roles in tumorigenesis [138,139]. For example, MCs are crucial to mediating tumor progression in C57BL/6-KitW-sh/W-sh mice upon injection of adenocarcinoma cells [104]. Moreover, pharmacological targeting of MCs by sodium cromoglycate treatment of mice with prostate cancer enhances tumor progression, advocating an environmental context in the modulation of the pro- or anti-tumor effects of MCs [107]. Similarly, this promoting function was further confirmed in a mouse model of pancreatic cancer, implying a correlation of MC accumulation within a tumor mass with cancer progression [140,141].

Furthermore, the loss of the neurofibromatosis type 1 (NF1) gene in MCs is implicated in neurofibroma development [139]. Preclinical studies utilizing animal models with impaired c-KIT function helped to gain a broader knowledge of the roles of MCs in tumor biology, and they provided proofs-of-concept that MCs can promote angiogenesis and metastasis, as well as the release of growth factors and enzymes [139,142,143]. Therefore, these findings hold promise for therapeutic approaches targeting the functions of MCs for precision chemotherapy or combinatorial therapies. Importantly, animal models will facilitate deciphering the detrimental or beneficial roles of MCs in cancer progression although mouse data will always require proper validation in clinical patient settings. Regrettably, other limitations apply, such as the fact that abnormalities do not only specifically arise from isolated MC functions but are, rather, a consequence of dynamic and complex interactions within the immune system. Additionally, another limitation is the lack of MC-specific inhibitors approved for clinical use [64,143]. However, therapeutics modulating the MC number and function, as well as the MC signal transduction and protection from the effects of the MC mediators, may provide a suitable path [59].

## 7. Mast Cells in Allergic Diseases

Originally, MCs were described as one of the main effector cells in allergic IgE-mediated responses [49]. MCs may also be important in non-allergic chronic lung diseases, including chronic obstructive pulmonary disease (COPD), idiopathic pulmonary fibrosis (IPF), and asthma [87]. The prevalence of allergic diseases such as asthma, allergic rhino-conjunctivitis, food allergies, as well as atopic dermatitis, which share similar risk factors and predispositions, is increasing worldwide [144]. The typical type-I allergic reactions are urticaria or, in the most severe cases, life-threatening anaphylaxis [145]. However, people suffering from allergic disorders can experience gentle to acute symptoms within minutes after exposure to eliciting allergens. Unfortunately, a repeated exposure to allergens may trigger long-term changes in the affected tissues through tissue remodeling [146], as, for example, in chronic T cell–driven inflammatory skin diseases, such as allergic contact dermatitis (ACD) [146,147].

IgE-mediated allergic inflammation is a complex process which begins with allergen uptake by professional APCs like immature skin dendritic cells. DCs then maturate as well as migrate to the regional lymph nodes, where they present antigens to naïve T cells to polarize differentiation of CD4+ T cells into IL-4-secreting Th2 subsets. Th2 subsets induce an isotype, switching from IgM to IgE class antibodies in B lymphocytes, which are released into the blood circulatory system to recognize the FcεRI receptors present on the circulating basophils as well as on tissue MCs [148]. The high-affinity binding of IgE by FcεRI activates several intracellular signaling pathways, especially the increased calcium influx, which triggers MCs degranulation. The release of preformed and granule-stored proinflammatory mediators during early phase reactions (Table 3) is mainly responsible for the clinical symptoms of type-1 allergies, also known as immediate hypersensitivity [149]. Furthermore, the mediators of the early-phase reactions contribute to late-phase reactions (Table 3), where MCs release numerous cytokines and chemokines, as well as growth factors [80]. Inflammatory cytokines activate local tissue cells, including endothelial, epithelial, smooth muscle cells, and neurons. Moreover, inflammatory granulocytic leukocytes (neutrophils, basophils, and eosinophils), as well as agranular leukocytes (monocytes and lymphocytes), migrate to the inflammation site [150]. Hence, the concerted action of preformed proinflammatory mediators and de novo synthesized mediators drive the allergic inflammation [45].

One of the most important early-phase mediators involved in IgE-mediated allergic reactions is histamine. MCs are recognized as the main source of this compound in the human body. Histamine can activate different APCs, including dendritic cells [153], but they can also act on different cell types of the innate and adaptive immunity through one of four different GPCRs, such as histamine receptors (Table 3). Histamine modulates the synthesis of adhesion and costimulatory molecules, as well as that of other signaling cytokines [60]. Moreover, IL-33, a member of the IL-1 family and a ligand for the heterodimeric ST2 receptor (encoded by the IL1RL1 gene), functions as an effector cytokine in later phases of the immune response. IL-33 can be produced by fibroblasts, MCs, dendritic cells, macrophages, osteoblasts, endothelial cells, and epithelial cells. The ST2 receptor is constitutively expressed on the surface of some innate immune cells like MCs, ILC2, Th2, and Treg cells. Signaling between IL-33 and the ST2 receptor triggers innate responses. Furthermore, IL-33 stimulates the production of Th2 marker cytokines such as IL-4 [68,155].

Non-IgE mediated mechanisms of MC activation and degranulation are triggered by neuropeptides, cytokines, and by membrane and intracellular MAS-related G protein-coupled receptor X-2, called MRGPRX2 in humans (the mouse analogue is known as Mrgprb2) [87]. The MRGPRX2 receptor drives innate immunity, wound healing, or neurogenic inflammation, thus causing pain as well as itching [152]. Furthermore, it is of key importance for immediate drug hypersensitivity (IDH), and thus offers therapeutic options for the use of MRGPRX2 antagonists to prevent or treat IDH [156]. The pharmacological manipulation of MCs may therefore be an important strategy in controlling the outcome of allergic, as well as inflammatory, responses. Considering the amount and diverse nature of pro-inflammatory mediators that MCs release, which affect most fundamental cellular processes, a selective treatment must act at multiple levels to inhibit the life-threatening reactions to allergens during anaphylaxis [146,157].

## 8. Mast Cells as Key Players in Microbial Commensalism and Pathogenesis

MCs constitute the first line of anti-pathogen defense due to their location and adequate mechanisms to act as sensors of local microbiota and the immune effectors against bacteria, viruses, parasites, fungi, and venom toxins (Table 4). Numerous studies on host–pathogen interplay provide compelling evidence that MCs fulfil beneficial functions against infectious diseases [158,159,160] although the scenarios are very complicated. The MC response to pathogenic challenges depends on the MC type, their anatomic tissue location, the MC-generated proteases, and mucosal (MMCs) or connective tissue MCs (CTMCs). MMCs reside between epithelial cells of mucosal tissues in the lung and intestine, whereas CTMCs are located in the intestinal submucosa, skin, peritoneal cavity, and surroundings of blood vessels [161,162].

Intestinal protection against nematodes, such as *Strongyloides ratti* and *S. venezuelensis*, is mediated by MMCs, as CTMCs are indispensable for terminating nematode infections [162,178]. MCs orchestrate type 2 immunity to helminths via the regulation of IL-25, IL-33, and thymic stromal lymphopoietin (TSLP) [162,178]. MCs secrete IL-4 and IL-13 in response to helminths and helminth-derived proteases. The MC-derived chymase mMCP1 limits *Trichinella spiralis* larvae in tissues, but it is not required for the expulsion of *Nippostrongylus brasilensis*. MCs are also activated by the parasite-specific antibody classes IgM, IgG, IgA, and IgE although there is limited evidence for the protective role of the latter one [3,179,180] while cell types such as ILCs or T cells may be a primary source of the Th2-driving IL-4 against Leishmania spp infection, MCs’ complex host-defense reactions against intracellular pathogens [169]. Generally, tissue-specific microbiota can influence the host immune status during the steady state and in dysbiosis [65]. For example, microbial metabolites may directly or indirectly activate MCs to secrete mediators and cytokines, including TNF-α, IL-1β, IL-4, IL-10, and IL-13 [17,27,181,182]. The MC-derived mediators increase tissue-resident macrophage crawling and migration towards infection sites. However, MCs can also express on their surface inhibitory molecules that inhibit macrophage homeostasis [16]. For instance, IL-4 impairs both macrophage phagocytosis and autophagy by altering signaling and decreased expression of the macrophage MARCO receptor [183]. MCs recruit migrating DCs and modify their amplitude to induce Th2 differentiation [184]. MCs are tuned by cytokines associated with inflammatory responses. For example, IL-3 is produced during nematode infections to regulate the FcεRI expression and to control the FcεRI-dependent secretory function of MCs [71]. MCs phagocytose C. albicans via TLR2/Dectin-1; then, the internalized fungi are eliminated by ROS, as well as by nitric oxide (NO) production [182]. Although the phagocytosis of fungi depends on their morphotypes and MC subtypes, hyphae are phagocytosed with much lower efficiency. Of note, MCs facing Candida spp. release anti-inflammatory cytokines such as IL-9. MCs can be tuned by IL-9 to promote immune tolerance or epithelial damage. IL-9 and MCs also contribute to barrier function loss, dissemination, and inflammation in patients with celiac diseases. Thus, C. albicans is able to exploit the versatility of the IL-9/MC axis for balancing between commensalism and pathogenesis [163,183]. In the early phase of invasive candidiasis, MCs release the neutrophil-attracting pro-inflammatory chemokine IL-8. MCs also secrete IL-16 and anti-inflammatory IL-1, and they produce the MC extracellular traps (MCETs) during late candidiasis, similar to NETs produced by neutrophils [185]. Although MCETs entrap without killing *Candida* [167], the NADPH oxidase-dependent MCETs are able to effectively limit infections by bacterial Streptococcus pyogenes [67].

Interestingly, the IL-9/MC axis may prevent the consequences of fungal colonization. Although MC-derived IL-9 affects the indoleamine-pyrrole 2,3-dioxygenase (IDO1) expression via the signal transducer and activator of transcription 3 (STAT3), MCs mediate immune suppression via tryptophan hydroxylase-1 (Tph-1), catalyzing the conversion of tryptophan to serotonin. CTMCs induce intestinal immune tolerance in general via Tph1 and IDO1 [163]. Of note, ingested yeasts can provide allergens in individuals sensitized to fungi, which results in multiple severe anaphylactic reactions [186]. MCs produce TGF-β when activated by IL-9 that stimulate lung fibrosis in invasive pulmonary or allergic aspergillosis [187]. The lipophilic skin commensal yeast *Malassezia sympodialis* can trigger MCs to release cysteinyl leukotrienes in a dose-dependent manner in both non- and IgE-synthesized BMMC. Thus, M. sympodialis alters IL-6 response via affecting IgE receptor cross-linked BMMCs [13] in patients with atopic eczema (AE), but not in healthy ones. Conversely, *M. sympodialis* stimulates IL-8 release from both healthy individuals and AE patients, following the IgE receptor cross-linking [177]. Further, the *Staphylococcus aureus* protein A and *Peptostreptococcus magnus* protein L bind to different domains of the FcεRI-bound IgE and act as the IgE superantigens [168]. 

MCs also play a role in the recruitment of inflammatory cells via the secretion of histamine and various inflammatory cytokines, such as TNF-α and IL-6, and CCL2 when facing *Mycobacterium tuberculosis* [151]. Remarkably, MCs attenuate pulmonary infections by *Klebsiella pneumoniae*, *Mycoplasma pneumoniae*, *Pseudomonas aeruginosa*, and *Hemophilus influenza* otitis, as well as E. coli peritoneal and urinary infections [185]. MCs also contribute to innate immune containment and recovery from various bacterial infections, such as acute bacterial peritonitis, middle ear infection by *H. influenza*, respiratory mycoplasma infection, as well as skin infections. Moreover, MCs display a pronounced tropism to viral infection sites, accompanied by substantial production of type I and II IFNs [166,185]. Hence, MCs constitute a major and under-appreciated local source of type I and III interferons, following viral challenge. However, MCs can also drive harmful inflammatory responses associated with viral infections, long-term fibrosis, and vascular leakage. Furthermore, MCs are activated by respiratory viruses, and induce acute bronchoconstriction and lung inflammation, as well as asthma pathogenesis [176,188]. Respiratory virus-activated MCs can cause either protection by fighting infection or deleterious effects by inducing inflammation. 

## 9. Immunotherapy Targeting Mast Cells

In order to determine how MCs operate in-vivo in a tissue-specific context, a possible way would be to develop drugs or antibodies to selectively ablate either all local MCs or only a subpopulation of interest. However, such a drug remains elusive—if it could exist at all. However, the therapeutic approach (Table 5) used in mastocytosis and other diseases where MC numbers are markedly increased exploits tyrosine kinase inhibitors (midostaurin, nilotinib, imatinib, and dasatinib) to target the c-KIT receptor action and MC tryptase inhibitors (gabexate mesylate, nafamostat mesylate, and tranilast) [189,190]. There is information on the efficacy of the midostaurin treatment in a patient with advanced systemic mastocytosis including MC leukemia (after Phase II clinical trials). For instance, imatinib mesylate enhances tumor growth along with the MC depletion in a murine model of breast carcinoma. Masitinib mesylate is approved for treatment of recurrent or unresectable grade III MC tumors in dogs in veterinary medicine [103]. Masitinib mesylate has been available in Europe since 2009 under the brand name Masivet. In the USA, it is distributed under the name Kinavet and has been available for veterinaries since 2011 [191]. Moreover, c-KIT-R tyrosine kinase inhibitors are used for evaluative clinical trials in humans in gastrointestinal stromal tumor (GIST), mastocytosis, and pancreatic cancer [143]. FDA-approved avapritinib (approval: Jan 9, 2020, brand name Ayvakit) is used in adults with unresectable or metastatic GIST who harbor a platelet-derived growth factor receptor alpha (PDGFRA) exon 18 mutation and for advanced and indolent systemic mastocytosis (in phase 2 development) [192,193]. The experimental drug Obatoclax (GX015-070), a novel BH3 mimetic, a small molecule drug, binds and blocks the antiapoptotic activity of several members of the Bcl-2 family; moreover, it induces growth arrest in primary human and canine neoplastic MCs, as well as in different MC lines [194]. Obatoclax exerts synergistic antineoplastic effects on MCs when combined with dasatinib [195]. An inhibitor of ribonucleotide reductase, gemcitabine is used in the treatment of various carcinomas [196]. Gemcitabine is a 2′-deoxycytidine having geminal fluoro-substituents in the 2′-position. Moreover, gemcitabine is used in combination with the TLR-2/6 agonist MALP-2 in patients with pancreatic carcinomas [197]. Furthermore, targeting the MC degranulation may inhibit tumor growth and neo-angiogenesis. Orantinib inhibits the phosphorylation of the stem cell factor receptor tyrosine kinase c-KIT, often expressed in acute myelogenous leukemia cells [198,199]. Sodium cromolyn inhibits mediators released from activated MCs [200]. Of note, hypoxia in tumors of cromolyn-treated mice is a direct consequence of MC stabilization. Inhibition of the MC degranulation results in a greater degree of blood clotting adjacent to the tumor, as well as an increase in hypoxia within the tumor. Thus, MCs play an important role in regulating blood clotting and hypoxia solid cancer. However, while sodium cromolyn enhances peri-tumoral blood clotting and intratumoral hypoxia [99], its interference with MC function remains controversial.

Finally, IL-37 emerged as a new anti-inflammatory cytokine which binds to the α-chain of the IL-18 receptor α (IL-18Rα) and downregulates TLR-MyD88 signaling in MCs. Inflammatory disorders are thereby regulated by the inhibition of nuclear factor-κB (NF-κB) and MAPK signaling [208]. Hence, IL-37 may be a new therapeutic cytokine for treating chronic inflammatory skin diseases such as psoriasis (PS), where MCs contribute to the over-expression of several proinflammatory cytokines, such as TNF and IL-1 family members [209]. Moreover, IL-37 was proposed as a new therapeutic tool in a chronic autoimmune inflammatory disease (SS) [210]. Additionally, the human cathelicidin LL-37 peptide found in MCs plays an essential role in tissue homeostasis and protects the host from tumorigenesis [211].

Nonetheless, more MC-modulating drugs are urgently required, and existing treatment strategies need a careful and meticulous reinvestigation to determine beneficial versus detrimental effects on the MC function in health and disease. Indeed, FDA-approved pharmacological agents that either block MC activity or drive the release of MC mediators, such as histamine, leukotrienes or TNF, may be useful to better manage chronic infections or MC-derived symptoms in cases of associated pathologies [212].

## 10. Concluding Remarks

The recognition of possible mechanisms underlying direct and indirect interactions of MCs with different immune and non-immune cells in healthy/pathological conditions has remained an unmet challenge. However, the post-genomic era has also facilitated the discovery of numerous, as-yet-unknown signaling molecules and cytokines that MCs exploit for controlling immune homeostasis. It is undisputed that MCs accumulate in injured and inflamed tissues, where they can amplify, suppress, or modulate the dynamic progression of inflammation, including septic response to infectious pathogens. Importantly, the signaling crosstalk of MCs with local tissue immunity can have beneficial functions, some of which hold promise for cancer immunotherapy, as well as for the treatment of infections. However, successful therapeutic targeting of MC functions also requires a better mechanistic understanding of the molecular principles shaping the detrimental roles MCs play in a tissue-specific manner. This is pivotal in avoiding unwanted short- as well as long-term toxicity when targeting MC roles in malignant diseases, infections, or autoimmunity. Thus, drugs that target MC functions in tissue-specific, as well as disease-specific contexts, are needed, but existing treatment strategies should also be reinvestigated to determine their potential effects on MCs.

## Figures and Tables

**Figure 1 ijms-23-02249-f001:**
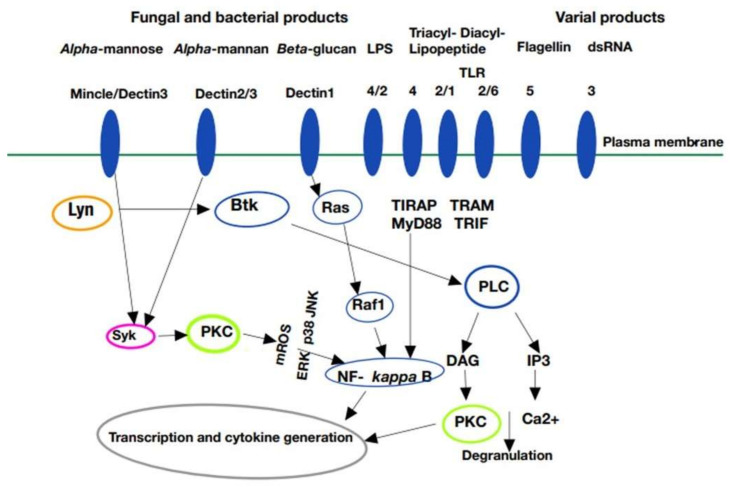
Pattern recognition receptors on mast cells and signal transduction pathways. Plasma membrane of the mast cell is a platform for pattern recognition receptors (PRRs), such as myeloid C-type lectin receptor (Dectin), Toll-like receptor (TLR), and macrophage-inducible C-type lectin (Mincle). Mincle/Dectin 3 (family member 3), Dectin 2/3, Dectin-1, and TLR: 4/2, 4, 2/1, 2/6, 5, 3 were identified respectively as receptors for: *alpha*-mannose, *alpha*-mannan, *beta*-glucan, lipopolysaccharide (LPS); triacyl-, diacyl-lipopeptide, flagellin, and double-stranded RNA (dsRNA). In a stressed mast cell (e.g., viral, bacterial, and fungal infections), activation of PRRs triggers the cell signaling, leading to the production of pro-inflammatory cytokines and degranulation. Mincle/Dectin3- and 2/3-dependent signaling activates tyrosine protein kinase Syk (known as spleen tyrosine kinase Syk) associated with protein kinase C (PKC) recruitment, resulting in mitochondrial reactive oxygen species (mROS) production and involvement of the following: c-Jun N-terminal kinase (JNK) and p38 mitogen-activated protein kinase (p38). Dectin 1 triggers the activity of a low-molecular-weight GDP/GTP-binding guanine triphosphatase (Ras); this, in turn, promotes a cascade of events leading to activation of proto-oncogene serine/threonine-kinase isoform Raf 1. Stimulation of the cell-surface receptors leads to activation of *kappa*-light-chain-enhancer nuclear factor of activated B cells (NF-*kappa* B) and rapid changes in gene expression. Intracellular signaling pathway involves tyrosine-protein kinase Lyn (Lyn) undergoing phosphorylation/activation and triggering a cascade of signaling events mediated by activation of Bruton’s tyrosine kinase (BTK) and phospholipase C (PLC) that results in the generation of second messengers: diacylglycerol (DAG) and inositol triphosphate (IP3), as well as in Ca^2+^ mobilization. DAG interacts with PKC, hence activating its influence on transcription and cytokine generation. Moreover, activation of NF-*kappa* B occurs through the intracellular docking proteins: myeloid differentiation factor 88 (MyD88) and Toll/Interleukin-1 (TIR)-containing adaptor inducing interferon-*beta* (TRIF). Thus, the signal is transduced through: MyD88, TIR-containing adaptor protein (TIRAP), and TRIF-related adaptor molecule (TRAM).

**Figure 2 ijms-23-02249-f002:**
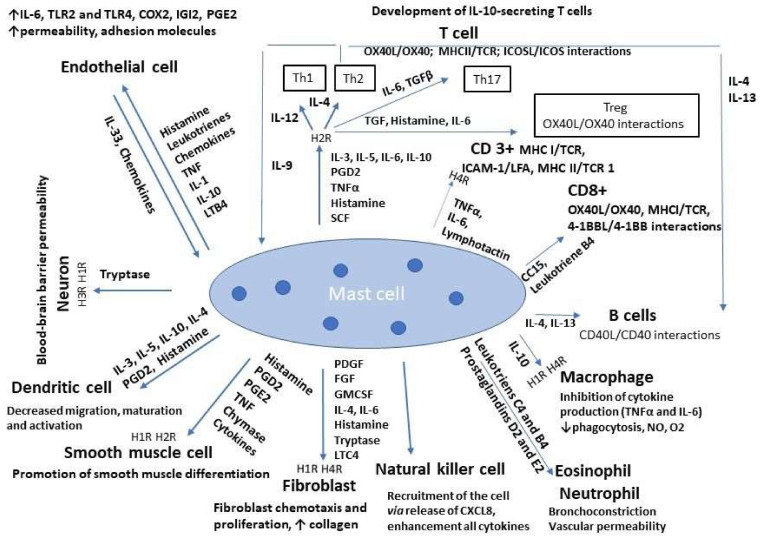
Mast cell-mediated modulation of cells of adaptive and innate immunity. Crosstalk of mast cells with other cells (dendritic, B and T, natural killer, fibroblasts, endothelium, smooth muscle, macrophages, and neurons) via the release of the following mediators: histamine, fibroblast 4 (FGF4), transforming growth factor (TGF-β), prostaglandin (PG) groups: D2 (PGD2) and E2 (PGE2), granulocyte-macrophage colony-stimulating factor (GMCSF), tryptase, leukotriene (LT) groups: C4 (LTC4) and B4 (LTB4), chymase and interleukin (IL) group (IL-1,-3,-4,-5,-6,-8,-10,-12,-13,-33), tumor necrosis factor (TNFα), stem cell factor (SCF), lymphotactin, protein reported as a synonym for the human gene VPREB1 (encoding V-set pre-B cell surrogate light chain 1-IGI2), nitric oxide (NO), oxide (O2), chemokine (C-X-C motif, ligand 8, and CXCL8), platelet-derived growth factor-4 (PDGF4), toll like receptors (TLR), cyclooxygenase (COX2), nerve growth factor (NGF), platelet-activating factor (PAF), substance P (SP), tumor necrosis factor α (TNF-α), and vascular endothelial growth factor (VEGF) family. Co-stimulation by reciprocal signals between MCs and lymphocytes involves the ligand–receptor interactions as well as released mediators. The ligand–receptor interactions are marked as follows: L’ refers to the ligand of a corresponding receptor; TNF superfamily member 4 (OX40; also known as TNFRSF4 or CD134), major histocompatibility complex (MHC), T-cell receptor (TCR), inducible co-stimulator (ICOS), cluster of differentiation 40 (CD40), TNF superfamily member 9 (4-1BB, also known as TNFRSF9 or CD137), Intercellular Adhesion Molecule 1 (ICAM-1, also known as CD54), and lymphocyte function-associated antigen 1 (LFA-1). Interaction of the MC-mediated histamine with other cell types includes endothelial cells and smooth muscle, neurons (H1R and H2R receptors), cells of the central nervous system (H3R), fibroblasts, macrophages, basophils, and eosinophils (H4R). Black up and down arrows represent the upregulation and downregulation.

**Table 1 ijms-23-02249-t001:** Crosstalk of mast cells with other cell types via the release of mediators.

	Cell Origin	MC-Released Mediators	References
Immune cells	Eosinophils	Histamine, IL-5, IL-9, SCF, LTD4, PAF, PGD2, VEGF-A	[9,53,55]
	Basophils	Histamine, PAF, PGD2	[9,35,56]
	Neutrophils	Histamine, LTB4, PAF, heparin	[7,8,9,35,49,53,54,57,58]
	Platelets	PAF	[58]
	Monocytes	Histamine, LTD4, VEGF-A, PAF	[9,53,55,59]
	Macrophages	Histamine, IL-13, IL-6, PAF, PGD2	[5,60,61,62,63]
	Dendritic cells	Histamine, PGE2, PGD2, VEGF-C, IL-13	[9,11,35,54,62,64,65,66]
	Innate lymphoid cells	IL-1β, IL-9, PGD2, LTD4	[59,67,68]
	NK cells	Histamine, heparin	[35,49,64]
	CD4^+^/CD8^+^ T cells	Histamine, LTC4, LTD4, TNF-α	[40,55,60]
	Th1 cells	Histamine	[11,60]
	Th2 cells	Histamine, LTC4, LTD4, PGD2	[9,11,35,54]
	Tfh cells	IL-6	[54]
	B cells	Histamine, PAF, IL-5	[35,58,63]
	Treg cells	Histamine	[40]
Non-immune cells	Blood endothelial cells	Histamine, LTC4, LTD4, PGD2, PAF, VEGF-A, IL-13, IL-1β	[55,69,70,71,72]
	Lymphatic endothelial cells	VEGF-C, VEGF-D	[73,74]
	Bronchial epithelia	IL-13, TNF-α, IL-9, TGF-β, PGD2	[31,55,56,75]
	Smooth-muscle cells	Histamine, LTC4, LTD4, PGD2, tryptase	[9,76,77,78,79]
	Goblet cells	IL-13, LTE4	[80]
	Cardiomyocytes	Tryptase	[31,54,72,81,82]
	Osteo-blasts/clasts	Histamine, osteopontin	[35,54,68,83,84,85,86]
	Neurons	Histamine, NGF, SP, PGD2	[9,53,85,86]
	Adipocytes	IL-13	[31,54,68,87]
	Fibroblasts	Tryptase, PGD2, TNF-α, TGF-β, IL-13, IL-1β	[9,53,55,60,88]
	Keratinocytes	Tryptase, PAF	[11,53]

**Table 2 ijms-23-02249-t002:** Pro-tumorigenic or anti-tumorigenic role of mast cells.

Pro-Tumorigenic Role of Mast Cells in Human Cancers	Pro-Tumorigenic Mechanisms	References
Bladder	Angiogenesis (VEGF-A, VEGF-B, FGF-2, IL-8), Genetic instability, DNA & RNA damage (ROS), Tumor cell growth (PAF), Immunosuppression (adenosine), Mesenchymal transition (beta TGF, IL-8), Tissue remodelling (Tryptase), Activation of STAT-3 (IL-6)	[29,50,54,55,56,62,63,69,93,94,95,96]
Colorectal	Angiogenesis (VEGF-A, VEGF-B, FGF-2, IL-8), Genetic instability, DNA & RNA damage (ROS), Tumor cell growth (PAF)	[64,95,97,98]
Esophagus	Angiogenesis (VEGF-A, VEGF-B, FGF-2, IL-8), Genetic instability, DNA & RNA damage (ROS), Tumor cell growth (PAF)	[55,56,62,63,78,99,100]
Gastric	Angiogenesis (VEGF-A, VEGF-B, FGF-2, IL-8), Genetic instability, DNA & RNA damage (ROS)	[9,49,100]
Hepatocellular	Angiogenesis (VEGF-A, VEGF-B, FGF-2, IL-8), Genetic instability, DNA & RNA damage (ROS), Tumor cell growth (PAF), Activation of STAT-3 (IL-6)	[55,101]
Hodgkin’s lymphoma	Angiogenesis (VEGF-A, VEGF-B, FGF-2, IL-8), Immunosuppression (adenosine), Mesenchymal transition (beta TGF, IL-8), IL-13 (Macrophage 2 polarization)	[99,102]
Melanoma	Angiogenesis (VEGF-A, VEGF-B, FGF-2, IL-8), Activation of STAT-3 (IL-6)	[9,55,60,62,63,76,103]
Pancreas	Angiogenesis (VEGF-A, VEGF-B, FGF-2, IL-8), Genetic instability, DNA & RNA damage (ROS), Tumor cell growth (PAF)	[55,62,70,74,98,104,105]
Prostate	Angiogenesis (VEGF-A, VEGF-B, FGF-2, IL-8), Genetic instability, DNA & RNA damage (ROS)	[94,100,106]
Thyroid	Tissue remodelling (Tryptase), Mesenchymal transition (beta TGF, IL-8), Genetic instability, DNA & RNA damage (ROS)	[55,96,107]
**Anti-tumorigenic Role of Mast Cells in Human Tumors**	**Antitumorigenic Role**	
Breast	Cytotoxicity (TNF alpha), Cytotoxicity (ROS), Tumor growth inhibition (heparin), Tumor cell inhibition (cytolytic activity), Tumor cell apoptosis (IL-4, TNF)	[58,59,62,100,102]
Colorectal	Cytotoxicity (TNF alpha), Cytotoxicity (ROS), Tumor growth inhibition (heparin), Tumor cell growth inhibition (IL-9)	[9,54,62]
Lung	Cytotoxicity (TNF alpha), Cytotoxicity (ROS), Tumor growth inhibition (heparin), Tumor cell apoptosis (IL-4, TNF)	[9,53,55,59,108]
Mesothelioma	Cytotoxicity (TNF alpha), Cytotoxicity (ROS), Tumor growth inhibition (heparin)	[55,109]
Melanoma	Cytotoxicity (TNF alpha), Cytotoxicity (ROS), Tumor growth inhibition (heparin), Tumor cell growth inhibition (IL-9), APC maturation (histamine), Tumor cell inhibition (cytolytic activity)	[9,55,60,62,103]
Ovarian	Cytotoxicity (TNF alpha), Cytotoxicity (ROS), Tumor growth inhibition (heparin),	[76,94,110,111]
Pancreas	Cytotoxicity (TNF alpha), Cytotoxicity (ROS), Tumor growth inhibition (heparin),	[55,62,70,98,105]
Prostate	Tumor cell apoptosis (IL-4, TNF), Immune cell recruitment and activation, Tumor cell growth inhibition (IL-9)	[94,100,106]
**Tumors in Which Mast Cells Play Both a Pro- and Anti-Tumorigenic Role**	**Pro-and Anti-Tumorigenic Mechanisms**	
Colorectal	Cytotoxicity (ROS), DNA & RNA damage (ROS)	[9,53,56,112]
Lung	Cytotoxicity (ROS), DNA & RNA damage (ROS), Genetic instability	
Prostate	DNA & RNA damage (ROS), Genetic instability	
Melanoma	DNA & RNA damage (ROS), Genetic instability	
Pancreatic	Cytotoxicity (ROS), DNA & RNA damage (ROS), Genetic instability	
**Non-Contributing Role of Mast Cells in Tumors**		
Colorectal	Mutations in *beta*-catenin or axin Inactivation of TGF-*beta* signaling pathway	[9,55,62,78]
Non-small cell lung cancer	Mutation in EGFR Mutation in anaplastic lymphoma kinase	[55,62,100,102]
Renal	Biallelic inactivation of *VHL*, Deletion of *SETD2, BAP1, PBRM1,* Constitutive activation of the HIF pathway (angiogenesis, anaerobic glycolysis, pentose phosphate pathway, epithelial–mesenchymal transition, invasion, and metastasis)	[113,114,115,116]

**Table 3 ijms-23-02249-t003:** Mediators secreted by mast cells during allergic reactions.

Mediators	MC-Interacting Cell Types	Biological Effects	References
Proteases Tryptase, MMPs Proteoglycans heparin Chemokines (MCP-1, RANTES, TARC Cytokines IL-1, IL-3, IL-4, IL-6, IL-18, IL-33, TNF-α, SCF, TGF-β Growth factors VEGF, bFGF, NGF, GM-CSF, M-CSF Lipid-derived LTC4, LTD4, LTE4, PGD2, PAF Neuropeptides CRH, VIP	Endothelial cells, Bronchial, Smooth muscle, Neurons, Vascular, Gastric parietal cells, Cells of the central nervous system, Basophils and eosinophils	Type-1 hypersensitivity, Pro-inflammatory effects, Wheal and flare reaction in skin, Bronchoconstriction, Drop in blood pressure, Gastrointestinal hyperactivity	[11,35,79,84,85,92,151]
Growth factors bFGF, VEGF, NGF Cytokines IL-1, IL-3, IL-4, IL-6, IL-18, IL-33, TNF-α, SCF, TGF-β	Vascular, Gastric parietal cells	Regulation of the immune response, Th1 lymphocyte cytokine production	[8,9,11,35,92,152]
Proteases tryptase, MMPs Neuropeptides CRH, VIP Lipid-derived LTC4, LTD4, LTE4, PGD2, PAF	Cells of the central nervous system, Neurons	Blood–brain barrier permeability	[11,40,41,82,125,129,153,154]
Chemokines MCP-1, RANTES, TARC Cytokines IL-1, IL-3, IL-4, IL-6, IL-18, IL-33, TNF-α, SCF, TGF-β	Basophils and eosinophils	Pro-inflammatory effects, Process of chemotaxis, Stimulation of histamine and cytokine generation	

**Table 4 ijms-23-02249-t004:** The role of mast cells in response to pathogenic and commensal microorganisms.

Biological Agents	Localization	MC Subpopulation	Signaling/Recruitment	References
**Pathogenesis**				
Bacteroidetes Proteobacteria	Gut	MMCs; MCPT1	IL-9, ILC-2; Th9	[163]
*Streptococcus equis*	Peritoneum	BMMC-derived MCs	IL-4, IL-6, IL-12, IL-13, TNF-α, chemokines (CCL2/MCP-1, CCL7/MCP-3, CXCL2/MIP-2, CCL5/RANTES)	[164]
*Haemophilus influenzae*	Middle ear	BMMC-derived MCs	-	[165]
*Mycoplasma pulmonis*	Lung	BMMC-derived MCs	TNF-α, MCP-1, MIP-2, IL-6, Histamine	[166]
*Pseudomonas aeruginosa*	Skin	BMMC-derived MCs	NF-α, MCP-1, MIP-2, and IL-6	[59]
*C. albicans* (yeast and hyphae)	Mucous membranes	BMMC-derived MCs, MMCs; MCPT1	IL-9, ILC-2; Th9, IL-4, IL-6, IL-8; IL-13, TNF-α; Neutrophils, Macrophage crawling and migration	[16,163,167]
*Malassezia sympodialis*	Skin	BMMC-derived MCs	IL-6; IL-8 β-hexosaminidase, IgE	[13]
*Staphylococcus aureus* *Peptostreptococcus magnus*	Skin	BMMC-derived MCs	IL-4; IL-13; IgE	[168]
*Strongyloides ratti and S. venezuelensisis*	Mucosal intestine	MMCs	IL-4, IgE	[162]
*Trichinella spiralis*	Mucosal intestine; vessels	BM-resident HPC	mMCP-1, IgE, IL-2,	[36]
*Leishmania major* *L. infantum*	Cutaneous tissues	BMMCs	Β-hexosaminidase, TNF-α, Chymases (mMCP-1, mMCP-9), NO, IgE; IL-12; IFN-γ	[169,170]
*Plasmodium berghei*	CLN; skin	MMCs	TNF-α; CCL2; CXCL1; MMP-9; IFN-γ	[171]
DENV	Mucous membranes	MMCs	Tryptase, IL-1β	[172]
Zika virus	Mucous membranes	MMCs	Histame, IL-9, Th2	[173,174]
Influenza and parainfluenza	Mucous membranes	MMCs	Th2, type I INF	[174]
HIV-1gp120	Vascularized tissues	MC progenitors in the blood	IgE	[175]
Hepatitis C	Vascularized tissues	MC progenitors in the blood	IL-10, type I TNF	[101]
Coronavirus	Mucous membranes of lungs	MMCs	Histamine, Protease, PGD2, LTC4, IL-1, IL-6, IL-33	[176]
**Homeostatic Conditions**				
*C. albicans*	Mucosal gut	CTMC	TGF-β; ILC2; IL-9; Th9; Treg Foxp3; IL-10	[163]
*C. albicans*	Mucous membrane	BMMC-derived MCs	Restrain myeloid cells	[16]
*Malassezia sympodialis*	Skin	BMMC-derived MCs	IL-8, IgE	[177]

**Table 5 ijms-23-02249-t005:** Drugs targeting MC-derived tumors.

Drug	Target	Mode of Action	References
Midostaurin Chemical name: N-[(2S,3R,4R,6R)-3-methoxy-2-methyl-16-oxo-29-oxa-1,7,17-triazaoctacyclo [1 2.12.2.12,6.07,28.08,13.015,19.020,27.021,26]nonacosa-8,10,12,14,19,21,23,25,27-nonaen-4-yl]-N-methylbenzamide Brand name: Rydapt	PKC alpha, VEGFR2, KIT	Antagonist and inhibitor	PubChem CID: 9829523 [103]
	WT and/or mutant FLT3 tyrosine kinases	Apoptosis of target leukemia cells expressing target receptors and mast cells, Antiproliferative activity, Interacts with OATP	
Imatinib Chemical name: 4-[(4-methylpiperazin-1-yl) methyl]-*N*-[4-methyl-3-[(4-pyridin-3-ylpyrimidin-2-yl)amino]phenyl]benzamide Brand name: Gleevec (USA) or Glivec (Europe/Australia)	BCRP	Inhibitor	PubChem CID: 5291 [64]
	MSCGFR KIT	Antagonist, multitarget	
	RET proto-oncogene	Inhibitor	
	HANGFR	Antagonist	
	PDGFRα	Antagonist	
	EDDCR 1	Antagonist	
	KIT ABL1	Inhibitor	
	Platelet-derived growth factor receptor beta	Antagonist	
Dasatinib Chemical name: *N*-(2-chloro-6-methylphenyl)-2-[[6-[4-(2-hydroxyethyl)piperazin-1-yl]-2-methylpyrimidin-4-yl]amino]-1,3-thiazole-5-carboxamide	KIT ABL1	Multitarget-bind to both the active and inactive conformation of the ABL kinase domain	PubChem CID: 3062316 [59,83]
	Proto-oncogene KIT Src	Multitarget	
	Ephrin type-A receptor 2	Antagonist	
	KIT Lck	Multitarget	
	KIT	Inhibitor	
	MSCGFR Kit	Antagonist	
	PDGFRP	Antagonist	
	STAT5B	Inhibitor	
	Abelson KIT2	Multitarget	
	KIT Fyn	Multitarget	
Nilotinib Chemical name: 4-methyl-*N*-[3-(4-methylimidazol-1-yl)-5-(trifluoromethyl)phenyl]-3-[(4-pyridin-3-ylpyrimidin-2-yl)amino]benzamide	KIT ABL1	Inhibitor of BCR–ABL-binds to the inactive conformation of ABL	PubChem CID: 644241 [201]
	MSCGFR Kit	Antagonist	
Tranilast Chemical name: 2-[[(E)-3-(3,4-dimethoxyphenyl)prop-2-enoyl]amino]benzoic acid Brand name: Rizaben	Hematopoietic prostaglandin D synthase	Inhibitor	PubChem CID: 5282230 [103,189]
	MAP kinases (extracellularly regulated kinase 1 and 2 and JNK)	Calcium channel blocker, Antineoplastic agent, Aryl hydrocarbon receptor agonist and a hepatoprotective agent, Inhibit TGF-β production, interferon-gamma, IL-6, IL-10, and IL-17 by lymphoid cells	
Gabexate Chemical name: 4-[[6-[(Aminoiminomethyl)amino]-1-oxohexyl]oxy]-benzoic acid ethyl ester mesylate Brand name: Gabexate mesylate (Japan), Foy (Taiwan), Reminaron (Japan)	Serine protease Thrombin Trypsin Plasminogen Plasma kallikrein	Inhibitor, Antithrombotic in vitro and in vivo, inhibits LPS-induced TNF-α production, Inhibiting NF-κB and AP-1 activation	PubChem CID: 3447 [202]
Nafamostat mesylate Chemical name: (6-carbamimidoylnaphthalen-2-yl) 4-(diaminomethylideneamino)benzoate Brand name: Fusan	Serine protease	Inhibits enzyme: thrombin, Xa, and XIIa), The kallikrein–kinin system, The complement system, Pancreatic proteases and activation of protease-activated receptors (PARs), Lipopolysaccharide-induced nitric oxide production, apoptosis, Interleukin (IL)-6 and IL-8. Antioxidant in TNF-α-induced ROS production	PubChem CID: 5311180 [203]
	Prothrombin Coagulation factor X Coagulation factor XII Trypsin-1 Kallikrein-1 Intracellular adhesion molecule 1	Inhibitor	
Masitinib mesylate Chemical name: N-(4-methyl-3-{[4-(pyridin-3-yl)-1,3-thiazol-2-yl]amino}phenyl)-4-[(4-methylpiperazin-1-yl)methyl]benzamide Brand name: Masivet, Kinavet	Tyrosine-kinase PDGF and KIT	Inhibiting the stem cell factor that regulates mast cell tumor proliferation, Antiproliferative actions- targets the c-KIT pathway	PubChem CID: 25024769 [103]
Avapritinib Chemical name: (1S)-1-(4-fluorophenyl)-1-(2-{4-[6-(1-methyl-1H-pyrazol-4-yl)pyrrolo[2,1-f][1,2,4]triazin-4-yl]piperazin-1-yl}pyrimidin-5-yl)ethan-1-amine Brand name: Ayvakit	MSCGFR KIT and PDGFR alpha	Inhibitor Negatively modulates the action of cell transporters	PubChem CID: 118023034 [204]
Obatoclax Chemical name: 2-[(2Z)-2-[(3,5-dimethyl-1H-pyrrol-2-yl)methylidene]-3-methoxy-2H-pyrrol-5-yl]-1H-indole	Apoptosis regulator Bcl-2	Leads to release of apoptosis-inducing cytochromec	PubChem CID: 11404337 [205]
	Bcl-2 family of proteins	Displaces BH3 domains by activation of a pocket of the BcL-2 family member	
Gemcitabine Chemical name: 4-amino-1-[(2R,4R,5R)-3,3-difluoro-4-hydroxy-5-(hydroxymethyl)oxolan-2-yl]-1,2-dihydropyrimidin-2-one Brand name: Gemzar, Infugem	Block DNA Replication DNA during the “S” phase (or DNA synthesis phase of the cell cycle), stopping normal development and division	Cross-linking/Alkylation	PubChem CID: 60750 [206]
	Ribonucleoside-diphosphate reductase large subunit	Blocks an enzyme which converts the cytosine nucleotide into the deoxy derivative	
	Thymidylate synthase	Blocks the incorporation of the thymidine nucleotide	
	UMP-CMP kinase	Inhibitor	
Orantinib Chemical name: 3-(2,4-dimethyl-5-{[(3Z)-2-oxo-2,3-dihydro-1H-indol-3-ylidene]methyl}-1H-pyrrol-3-yl)propanoic acid	KIT autophosphorylation VEGFR2 PDGFR FGFR angiogenesis proliferation	Inhibitor	PubChem CID: 5329099 [207]
Cromolyn Chemical name: 5-{3-[(2-carboxy-4-oxo-4H-chromen-5-yl)oxy]-2-hydroxypropoxy}-4-oxo-4H-chromene-2-carboxylic acid Brand name: Spinhaler, nebulizer solution, metered-dose inhaler, NasalCrom, Nasal Allergy Control, Nasalcrom Child	Protein S100-P	Antagonist	PubChem CID: 2882 [99,200]
	Calcium sensor Capsaicin receptor	Ion channel blockade Blockade	
	Cellular calcium signaling	Blockade of signaling of heat shock protein or G-protein	
	Stabilizing mast cells	Prevents the subsequent release of mediators; Precise mechanism/clinical activity not delineated	

## Data Availability

No new data were created or analyzed in this study. Data sharing is not applicable to this article.

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
