# Peer review of "The Multifaceted Roles of Mast Cells in Immune Homeostasis, Infections and Cancers"

_ijms, 2022, doi:10.3390/ijms23042249_

Round 1

Reviewer 1 Report

This review demonstrates the role of mast cells in the homeostasis of immune system. Generally, this manuscript is well organized and provides sufficient information about mast cells in the modulation of immune responses to cancers and infections. However, there are several minor points should be noticed. 

Minor points:

  1. line 54 to 56, the authors mention that autoimmune diseases including atherosclerosis, allograft rejection, neoplasia, rheumatoid arthritis. However, among these diseases, only rheumatoid arthritis belongs to autoimmune disease. Therefore, this sentence should be rephrased.
  2. Authors also demonstrate the promoting or inhibitory effects of mast cells in different cancers, thus, this part should be included in the title of this manuscript, too.
  3. The underline in the words, such as line 116-117 and table 2 and 5, should be removed. 
  4. Line 574-576, ref. 85 did not demonstrate the role of mast cells in the production of proinflammatory cytokines in psoriasis, therefore, authors should cite paper which describe the role of mast cells in the psoriasis.
  5. In the paragraph discuss about IL-37, several novel studies demonstrated about the association between mast cells and IL-37, such as Int J Mol Sci. 2020 Jun 16;21(12):4297, Int J Mol Sci. 2021 Jul 28;22(15):8076, Int Immunopharmacol. 2020 Jun;83:106476, Crit Rev Immunol. 2019;39(4):267-274, should be included in this paragraph.

Author Response

Dear Reviewer,

Thank you for all comments. Below you will find the Authors' replay to Editor's and Reviewers' comments. 

Dear Liliana Carmen Tatar
Assigned Editor,

Thank you for the comments. We have gone over the entire revision and corrected the writing style where applicable. The Authors’ correction according to the Editor’s and Reviewers’ comments was included in the revised manuscript (marked in yellow in Track Changes using MS Word). Authors’ answers to Editor's and Reviewers’ comments are included below in the original letter form the Editor (marked in black in Table below). The suitable letter was prepared below and the Author’s answers are indicated in red. Additional changes introduced by the Authors were marked in yellow. English (British spelling was used) and the style of manuscript was corrected and marked in the proofreading program in the original Ms.

Regards,

Monika Staniszewska

Editor's original letter:

(I)  Please revise your manuscript according to the referees’ comments and upload the revised file within 5 days.

(II) Please use the version of your manuscript found at the above link for your revisions.
(III) Any revisions made to the manuscript should be marked up using the “Track Changes” function if you are using MS Word/LaTeX, such that changes can be easily viewed by the editors and reviewers.
(IV) Please provide a short cover letter detailing your changes for the editors’ and referees’ approval.

If one of the referees has suggested that your manuscript should undergo extensive English revisions, please address this issue during revision. We propose that you use one of the editing services listed at
https://www.mdpi.com/authors/english or have your manuscript checked by a native English-speaking colleague.
Please do not hesitate to contact us if you have any questions regarding the revision of your manuscript or if you need more time. We look forward to hearing from you soon.

Rev#1

This review demonstrates the role of mast cells in the homeostasis of immune system. Generally, this manuscript is well organized and provides sufficient information about mast cells in the modulation of immune responses to cancers and infections. However, there are several minor points should be noticed. 

Minor points:

Rev#1’s comments

Authors’ answers

  1. line 54 to 56, the authors mention that autoimmune diseases including atherosclerosis, allograft rejection, neoplasia, rheumatoid arthritis. However, among these diseases, only rheumatoid arthritis belongs to autoimmune disease. Therefore, this sentence should be rephrased.

The sentence was rephrased as follows:

In autoimmune diseases (rheumatoid arthritis), chemokines (CXCL12, CCL2, CCL3, CCL4, CCL5) are implicated in tissue destruction [5–7].

2.     Authors also demonstrate the promoting or inhibitory effects of mast cells in different cancers, thus, this part should be included in the title of this manuscript, too.

Title was modified as follows:

The Multifaceted Roles of Mast Cells in Immune Homeostasis, Infections and Cancers

3.     The underline in the words, such as line 116-117 and table 2 and 5, should be removed. 

Lines were removed in:

Lines 116-117: C-type lectin receptor (Dectin),

Table 2: anaplastic lymphoma kinase

Table 5: RET proto-oncogene, Proto-oncogene KIT Src, Ephrin type-A receptor 2, KIT Lck, Abelson KIT2, KIT, Hematopoietic prostaglandin D synthase, Apoptosis regulator Bcl-2, Ribonucleoside-diphosphate reductase large subunit, Thymidylate synthase, UMP-CMP kinase, Protein S100-P

4.     Line 574-576, ref. 85 did not demonstrate the role of mast cells in the production of proinflammatory cytokines in psoriasis, therefore, authors should cite paper which describe the role of mast cells in the psoriasis.

Ref. was corrected: Hence, IL‐37 may be a new therapeutic cytokine for treating chronic inflammatory skin diseases such as psoriasis (PS), where MCs contribute to the over-expression of several proinflammatory cytokines such as TNF and IL-1 family members [229].

5.     In the paragraph discuss about IL-37, several novel studies demonstrated about the association between mast cells and IL-37, such as

Int J Mol Sci. 2020 Jun 16;21(12):4297, Int J Mol Sci. 2021 Jul 28;22(15):8076, Int Immunopharmacol. 2020 Jun;83:106476,

Crit Rev Immunol. 2019;39(4):267-274, should be included in this paragraph.

Quotations and the following discussion was added: Inflammatory disorders are thereby regulated by the inhibition of nuclear factor‐κB (NF‐κB) and MAPK signalling [228]. Hence, IL‐37 may be a new therapeutic cytokine for treating chronic inflammatory skin diseases such as psoriasis (PS), where MCs contribute to the over-expression of several proinflammatory cytokines such as TNF and IL-1 family members [229]. Moreover, IL-37 was proposed as a new therapeutic tool in a chronic autoimmune inflammatory disease (SS) [230]. Additionally, the human cathelicidin LL-37 peptide found in MCs plays an essential role in tissue homeostasis and protects the host from tumorigenesis [231].

Rev2

This article reviews the diverse roles of mast cells. Mast cells are long-lived, tissue-resident cells, and the authors specifically discuss their role in immune homeostasis and in host defense against pathogens. The authors summarize the multifaceted nature of mast cells, which can contribute to host health, but also have negative effects on host tissues, in the following sections.

The authors summarize the types of Pattern Recognition Receptors that mast cells possess, their ligands, and the signaling pathways within mast cells after recognition. These responses result in pathogen elimination in the case of a moderate inflammatory response. On the other hand, mast cells are involved in tissue destruction in many inflammatory diseases because they release chemokines that recruit leukocyte subtypes that are harmful to the host tissue.

The synthesis process of secretory granules of mast cells, their contents, and their effects are described.

The authors also review in detail the effects of cytokines, chemokines, and secretory granules released from activated mast cells on non-mast immune cells and tissue cells.

While mast cells exhibit anticancer activity, they can also promote tumor invasion by inducing angiogenesis.

 This paper introduces the dynamics and functions of mast cells, which could be analyzed by the existence of a mast cell-deficient mouse model.

The multifaceted mast cell actions are summarized by drawing on a vast amount of literature, but much of the information has already been described in other reviews.

However, it is a carefully written review and the mention of mast cell-neuron interaction is new and quite interesting.

A few minor revisions are listed below.

Rev#2’s comments

Authors’ answers

There is a typographical error in p15, line 493; IL-1ra. 

It was corrected as follows: MCs also secrete IL-16 and anti-inflammatory IL-1, and produce the MC extracellular traps (MCETs) during late candidiasis, similar to NETs produced by neutrophils [135].

Reviewer 2 Report

This article reviews the diverse roles of mast cells. Mast cells are long-lived, tissue-resident cells, and the authors specifically discuss their role in immune homeostasis and in host defense against pathogens. The authors summarize the multifaceted nature of mast cells, which can contribute to host health, but also have negative effects on host tissues, in the following sections.

 The authors summarize the types of Pattern Recognition Receptors that mast cells possess, their ligands, and the signaling pathways within mast cells after recognition. These responses result in pathogen elimination in the case of a moderate inflammatory response. On the other hand, mast cells are involved in tissue destruction in many inflammatory diseases because they release chemokines that recruit leukocyte subtypes that are harmful to the host tissue.

The synthesis process of secretory granules of mast cells, their contents, and their effects are described.

The authors also review in detail the effects of cytokines, chemokines, and secretory granules released from activated mast cells on non-mast immune cells and tissue cells.

 While mast cells exhibit anticancer activity, they can also promote tumor invasion by inducing angiogenesis.

 This paper introduces the dynamics and functions of mast cells, which could be analyzed by the existence of a mast cell-deficient mouse model.

The multifaceted mast cell actions are summarized by drawing on a vast amount of literature, but much of the information has already been described in other reviews.

However, it is a carefully written review and the mention of mast cell-neuron interaction is new and quite interesting.

A few minor revisions are listed below.

There is a typographical error in p15, line 493; IL-1ra. 

Author Response

Dear Reviewer,

Thank you for all comments. Below you will find the authors' answers to Editor's and Reviewers' comments. 

Dear Liliana Carmen Tatar
Assigned Editor,

Thank you for the comments. We have gone over the entire revision and corrected the writing style where applicable. The Authors’ correction according to the Editor’s and Reviewers’ comments was included in the revised manuscript (marked in yellow in Track Changes using MS Word). Authors’ answers to Editor's and Reviewers’ comments are included below in the original letter form the Editor (marked in black in Table below). The suitable letter was prepared below and the Author’s answers are indicated in red. Additional changes introduced by the Authors were marked in yellow. English (British spelling was used) and the style of manuscript was corrected and marked in the proofreading program in the original Ms.

Regards,

Monika Staniszewska

Editor's original letter:

(I)  Please revise your manuscript according to the referees’ comments and upload the revised file within 5 days.

(II) Please use the version of your manuscript found at the above link for your revisions.
(III) Any revisions made to the manuscript should be marked up using the “Track Changes” function if you are using MS Word/LaTeX, such that changes can be easily viewed by the editors and reviewers.
(IV) Please provide a short cover letter detailing your changes for the editors’ and referees’ approval.

If one of the referees has suggested that your manuscript should undergo extensive English revisions, please address this issue during revision. We propose that you use one of the editing services listed at
https://www.mdpi.com/authors/english or have your manuscript checked by a native English-speaking colleague.
Please do not hesitate to contact us if you have any questions regarding the revision of your manuscript or if you need more time. We look forward to hearing from you soon.

Rev#1

This review demonstrates the role of mast cells in the homeostasis of immune system. Generally, this manuscript is well organized and provides sufficient information about mast cells in the modulation of immune responses to cancers and infections. However, there are several minor points should be noticed. 

Minor points:

Rev#1’s comments

Authors’ answers

  1. line 54 to 56, the authors mention that autoimmune diseases including atherosclerosis, allograft rejection, neoplasia, rheumatoid arthritis. However, among these diseases, only rheumatoid arthritis belongs to autoimmune disease. Therefore, this sentence should be rephrased.

The sentence was rephrased as follows:

In autoimmune diseases (rheumatoid arthritis), chemokines (CXCL12, CCL2, CCL3, CCL4, CCL5) are implicated in tissue destruction [5–7].

2.     Authors also demonstrate the promoting or inhibitory effects of mast cells in different cancers, thus, this part should be included in the title of this manuscript, too.

Title was modified as follows:

The Multifaceted Roles of Mast Cells in Immune Homeostasis, Infections and Cancers

3.     The underline in the words, such as line 116-117 and table 2 and 5, should be removed. 

Lines were removed in:

Lines 116-117: C-type lectin receptor (Dectin),

Table 2: anaplastic lymphoma kinase

Table 5: RET proto-oncogene, Proto-oncogene KIT Src, Ephrin type-A receptor 2, KIT Lck, Abelson KIT2, KIT, Hematopoietic prostaglandin D synthase, Apoptosis regulator Bcl-2, Ribonucleoside-diphosphate reductase large subunit, Thymidylate synthase, UMP-CMP kinase, Protein S100-P

4.     Line 574-576, ref. 85 did not demonstrate the role of mast cells in the production of proinflammatory cytokines in psoriasis, therefore, authors should cite paper which describe the role of mast cells in the psoriasis.

Ref. was corrected: Hence, IL‐37 may be a new therapeutic cytokine for treating chronic inflammatory skin diseases such as psoriasis (PS), where MCs contribute to the over-expression of several proinflammatory cytokines such as TNF and IL-1 family members [229].

5.     In the paragraph discuss about IL-37, several novel studies demonstrated about the association between mast cells and IL-37, such as

Int J Mol Sci. 2020 Jun 16;21(12):4297, Int J Mol Sci. 2021 Jul 28;22(15):8076, Int Immunopharmacol. 2020 Jun;83:106476,

Crit Rev Immunol. 2019;39(4):267-274, should be included in this paragraph.

Quotations and the following discussion was added: Inflammatory disorders are thereby regulated by the inhibition of nuclear factor‐κB (NF‐κB) and MAPK signalling [228]. Hence, IL‐37 may be a new therapeutic cytokine for treating chronic inflammatory skin diseases such as psoriasis (PS), where MCs contribute to the over-expression of several proinflammatory cytokines such as TNF and IL-1 family members [229]. Moreover, IL-37 was proposed as a new therapeutic tool in a chronic autoimmune inflammatory disease (SS) [230]. Additionally, the human cathelicidin LL-37 peptide found in MCs plays an essential role in tissue homeostasis and protects the host from tumorigenesis [231].

Rev2

This article reviews the diverse roles of mast cells. Mast cells are long-lived, tissue-resident cells, and the authors specifically discuss their role in immune homeostasis and in host defense against pathogens. The authors summarize the multifaceted nature of mast cells, which can contribute to host health, but also have negative effects on host tissues, in the following sections.

The authors summarize the types of Pattern Recognition Receptors that mast cells possess, their ligands, and the signaling pathways within mast cells after recognition. These responses result in pathogen elimination in the case of a moderate inflammatory response. On the other hand, mast cells are involved in tissue destruction in many inflammatory diseases because they release chemokines that recruit leukocyte subtypes that are harmful to the host tissue.

The synthesis process of secretory granules of mast cells, their contents, and their effects are described.

The authors also review in detail the effects of cytokines, chemokines, and secretory granules released from activated mast cells on non-mast immune cells and tissue cells.

While mast cells exhibit anticancer activity, they can also promote tumor invasion by inducing angiogenesis.

 This paper introduces the dynamics and functions of mast cells, which could be analyzed by the existence of a mast cell-deficient mouse model.

The multifaceted mast cell actions are summarized by drawing on a vast amount of literature, but much of the information has already been described in other reviews.

However, it is a carefully written review and the mention of mast cell-neuron interaction is new and quite interesting.

A few minor revisions are listed below.

Rev#2’s comments

Authors’ answers

There is a typographical error in p15, line 493; IL-1ra. 

It was corrected as follows: MCs also secrete IL-16 and anti-inflammatory IL-1, and produce the MC extracellular traps (MCETs) during late candidiasis, similar to NETs produced by neutrophils [135].